# Semantic Pruning for Single Class Interpretability

## Abstract

Convolutional Neural Networks (CNN) have achieved state-of-the-art performance in different computer vision tasks, but at a price of being computationally and power intensive. At the same time, only a few attempts were made toward a deeper understanding of CNNs. In this work, we propose to use semantic pruning technique toward not only CNN optimization but also as a way toward getting some insight information on convolutional filters correlation and interference. We start with a pre-trained network and prune it until it behaves as a single class classifier for a selected class. Unlike the more traditional approaches which apply retraining to the pruned CNN, the proposed *semantic pruning* does not use retraining. Conducted experiments showed that a) for each class there is a pruning ration which allows removing filters with either an increase or no loss of classification accuracy, b) pruning can improve the interference between filters used for classification of different classes c) effect between classification accuracy and correlation between pruned filters groups specific for different classes.

## 1 Introduction

Convolutional Neural Networks (CNN) provides high quality features that are used in various tasks such as object recognition (Ouyang et al., 2015; Wang et al., 2015; Lin et al., 2017a; Hu et al., 2017), scene classification (Zhou et al., 2014; Ren & Li, 2015), semantic segmentation Girshick et al. (2014); Husain et al. (2017), image captioning (Anderson et al., 2017; Rennie et al., 2017; Yao et al., 2017; Lu et al., 2017), raw audio generation (van den Oord et al., 2016), data analytics (Najafabadi et al., 2015). Unlike engineered features, where one or multiple features have been designed with specific domain knowledge (SIFT (Lowe, 1999) ,SURF (Bay et al., 2008), LBP (Ojala et al., 1994), Canny (Canny, 1986), etc.), features in CNN are obtained by directly learning representations (filters) from the data without explicit domain expertise.

While using a large number of learned filters in CNNs creates high-quality features, it also requires significant computational resources. The filter learning and the inference processes need resources such as GPU or TPUs to tackle the computational overhead. Recently, multiple optimization methods were proposed, with the target to reduce the computational overhead of either the learning or the inference processes. For instance, in the learning optimization approach, the following works have been done (Zhang et al., 2018; Lin et al., 2017b). On the other hand, the number of the current works targeting the optimization of the inference are: reduced precision computation (Rastegari et al., 2016; Courbariaux et al., 2016; Zhou et al., 2016), look-up and precomputing (Abdiyeva et al., 2018; Brendel & Bethge, 2019), pruning (Anwar et al., 2015; Li et al., 2016; Zhu & Gupta, 2017; Raghu et al., 2017; Morcos et al., 2018; Ma et al., 2019) and knowledge distillation (Hinton et al., 2015).

The work described in this paper continues in the direction of inference optimization by pruning. However, the pruning is only used as a tool for studying the interpretability of CNNs filter. Pruning is the process of removing redundant filters from the network (Anwar et al., 2015; Li et al., 2016; Zhang et al., 2017; Zylberberg, 2017; Zhu & Gupta, 2017; Raghu et al., 2017; Morcos et al., 2018; Ma et al., 2019). In current approaches, the pruning process consists of the following steps: train CNN, remove redundant filters and retrains the network to preserve the original accuracy. The proposed pruning method, however, does not involve retraining of the pruned network. The motivation of this work is the interpretation of CNNs: pruned and unpruned filters are analyzed to evaluate

class-wise relations and dependencies. In particular, the pruning applied to AlexNet is used to explore and study whether one can obtain a new single class classifier by dropping the selected filters in the original multi-class classifier instead of an expensive training process of a new classifier.

The interpretability of the network resulting from pruning can be seen as a filter-wise and signal-wise network dissection: The removal of filters in a controlled manner allows us to determine how different classified objects affects other classes, how the density of the classes in the learned feature space affects the accuracy of a classification and how the structure of the learned multi-class feature space allows to improve the accuracy of a single-class classification. Therefore, the pruning used in this work can also be seen as an approach to CNN understanding (Bau et al., 2017; Zylberberg, 2017; Kindermans et al., 2017; Morcos et al., 2018).

The proposed method, referred to as *semantic pruning*, is performed as follows: we start from a CNN trained to distinguish $q$ classes. Then we reduce the multi-class CNN to the single class classification problem, by removing unrelated to the selected class labels. To remove filters, we measure each filter's contribution (activation magnitude) toward recognition of a target object class. Filters with the highest contributions were preserved, and the remaining filters were dropped from the network. The pruned and unpruned filters are analyzed for correlation, overlap and other measures allowing us to determine class-wise interactions.

The contribution of this paper can be summarized as the following:

- investigated the sensitivity of object classification to semantic pruning,
- study of the interference between class wise filters in CNN,

This paper is organized as follows. Section 2 introduces semantic pruning concept. Section 3 describes the experiments and the results and Section 4 concludes the findings.

## 2 PROPOSED SEMANTIC PRUNING

Let $\mathbf{F}_i \in \mathbb{R}^{M \times M \times D}$ be a filter in layer $j$ of a convolutional network. Let, $\mathbf{B}_j \in \mathbb{R}^{X \times Y \times D}$ be an input tensor to the layer $j$. $X$ and $Y$ are spatial dimensions. To calculate the output tensor $\mathbf{B}_{j+1}$ of the layer $j$ input tensor $\mathbf{B}_j$ is convolved with the set of filters $\mathbb{L} = \{\mathbf{F}_1 ..., \mathbf{F}_{D'}\}$ of layer $j$. Lets denote the output of the convolution operation between input tensor $\mathbf{B}_j$ and a single filter $\mathbf{F}_i$ as $\boldsymbol{R}_i$:

$$\boldsymbol{R}_i = \mathbf{F}_i \odot \mathbf{B}_j \tag{1}$$

Then $\mathbf{B}^{j+1}_{(:,:,i)} = \boldsymbol{R}_i$

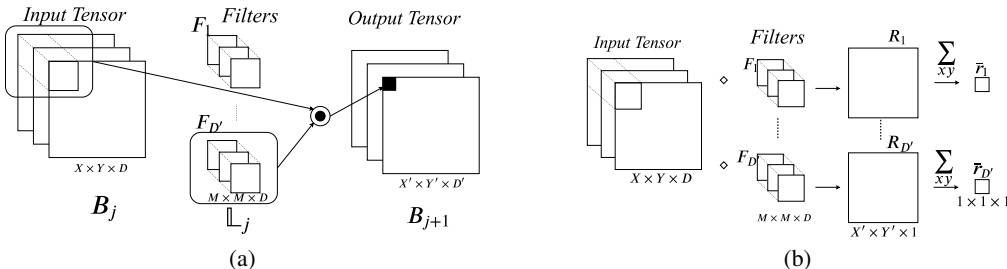

Figure 1: (a) Convolutional layer operation and (b) computation of the filter Activation Magnitude.

**Definition 1** (Pruning). *Let* $\mathbb{L} = \{\mathbf{F}_1, \ldots, \mathbf{F}_{D'}\}$ *be a convolutional layer,* $\mathbf{F}_i \in \mathbb{R}^{M \times M \times D}$ *be a single filter of the layer with an input tensor* $\mathbf{B}_j \in \mathbb{R}^{X \times Y \times D}$. $\mathbf{B}_{j+1} = \mathbb{L} \odot \mathbf{B}_j$ *(eq. 1) with* $\mathbf{B}_{j+1} \in \mathbb{R}^{X' \times Y' \times D'}$. *Pruning layer* $\mathbb{L}$ *denoted as* $\Omega(\mathbb{L})$ *is the process of removing one or more* $\mathbf{F}_i$ *from* $\mathbb{L}$ *resulting in a new set of filters* $\Omega(\mathbb{L}) = \mathbb{L}' = \{\mathbf{F}_1, \ldots, \mathbf{F}_{D''}\}$. *Hence, a new output is* $\mathbf{B}'_{j+1} = \mathbb{L}' \odot \mathbf{B}_j$ *with* $\mathbf{B}'_{j+1} \in \mathbb{R}^{X' \times Y' \times D''}$ *such that* $D'' \leq D'$.

In this work, we propose a semantic pruning method that is applied with respect to each selected class separately. We determine the amount of pruned filters for each object class $\lambda_i \in \Lambda =$

$\{\lambda_1, \ldots, \lambda_q\}$ where $q$ is the number of classes in the dataset. To determine which filters can be removed, the process of pruning requires a measure of contribution. In this work, we propose to use activation magnitude as filter removal criteria.

**Definition 2** (Activation Magnitude $\bar{r}$). *Let $\boldsymbol{R} \in \mathbb{R}^{X' \times Y'}$ be a result of convolution between input tensor and a single filter in the layer, then activation magnitude for each filter $\mathbf{F}_i$ is computed as follows:*

$$\bar{r}_i = \sum_{x=1}^{X'} \sum_{y=1}^{Y'} \mathbf{R}_{x,y,i} \tag{2}$$

*with $\bar{\boldsymbol{r}} \in \mathbb{R}^{1 \times 1 \times D'}$.*

The schema for calculating $\bar{r}$ is shown in Figure 1b.

We propose to perform pruning for each class separately; hence, we split the dataset into subsets of images belonging to the same class. Let $\mathbb{D} = \{\delta_1, \ldots, \delta_n\}$ be the dataset containing the $n$ data samples. Let $\Lambda = \{\lambda_1, \ldots, \lambda_q\}$ be the number of object classes in the dataset. Then, $\mathbb{D}_{\lambda_c} \subset \mathbb{D}$ be a subset of $\mathbb{D}$ containing all data samples with class label $\lambda_c$. The number of samples in $\mathbb{D}_{\lambda_c}$ is denoted as $p$, such that $p \leq n$. Let $\bar{r}_i(\delta_o)$ be the activation magnitude of a filter $\mathbf{F}_i$ resulting from using input sample $\delta_o \in \mathbb{D}_{\lambda_c}$. We want to measure the response of each filter w.r.t. to the individual class. Therefore, we propose to accumulate the activation magnitudes of all samples belonging to the selected class. The resultant value is denoted as *Accumulated Response*.

**Definition 3** (Accumulated Response $\gamma$). *$\bar{r}_i^{\lambda_c}$ is the Accumulated Response for filter $\mathbf{F}_i$ and object class $\lambda_c$ and defined as shown in eq. 3*

$$\gamma = \bar{r}_i^{\lambda_c} = \sum_{o=1}^{p} \bar{r}_i(\delta_o) \tag{3}$$

The process of semantic pruning starts by separating the dataset $\mathbb{D}$ into subsets $\mathbb{D}_{\lambda_c} = \{\delta_1, \ldots, \delta_p\}$ with $p$ being the number of samples in the subset $\mathbb{D}_{\lambda_i}$, for object label $\lambda_c \in \Lambda = \{\lambda_1, \ldots, \lambda_q\}$ with $q$ being the number of object classes. Let $m$ be the total number of filters in AlexNet. We calculate the accumulated response $\bar{r}^{\lambda_c} = \{\bar{r}_1^{\lambda_c}, \ldots, \bar{r}_m^{\lambda_c}\}$ for each filter in the network and for selected number of object classes from the dataset. The percentage of pruned filters is represented by the parameter $\theta$ (*pruning ratio*) such that $\theta = [0, \ldots, 1]$. For instance, for $\theta = 0.1$, the 10% of filters with lowest accumulated response will be pruned. In this work, let $\mathbf{F}_\sigma$ and $\mathbf{F}_{\bar{\sigma}}$ be the set of unpruned and pruned filters for a given $\gamma$ respectively. The process of filter pruning can be performed using two different approaches: network wise and layer-wise. Network wise pruning (NWP) is performed on filters $\mathbf{F}$ from all layers of the network. Filters are sorted according to their corresponding accumulated responses, and the filters with the smallest values are removed in correspondence to the pruning ratio $\theta$. On the other hand, Layer wise pruning (LWP) separates filters into groups, each containing filters from one layer. The pruning is then applied to each layer separately. However, as a result of removing certain filters, the activation response for the filters in the consecutive layers can get affected, and therefore, higher potential of pruning can be achieved. Therefore, we also provided a comparison between these two cases, referred to as Direct and Residual Pruning.

Two settings were used toward varying-parameter $\theta$. These approaches will be referred to as discrete and continuous pruning. For discrete pruning, the values are changes with the fixed step size, while for continues pruning the wider range of values is explored.

## 3 EXPERIMENTS

To illustrate the effectiveness of the semantic pruning approach, we applied the method on the model trained for an object classification task. In our experiments we used AlexNet (Krizhevsky et al., 2012) trained on ImageNet (Russakovsky et al., 2015) dataset. The AlexNet network consists of five convolutional and three fully connected layers. Fully connected layers are referred to as *classifier*. The convolutional layers are referred to as *feature extractor*.

The ImageNet data set contains 1000 object classes; however, due to computational constraints, we evaluated our method on randomly chosen 50 classes, to show the further potential of the method.

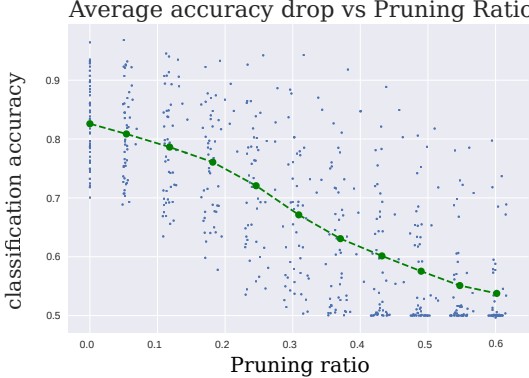
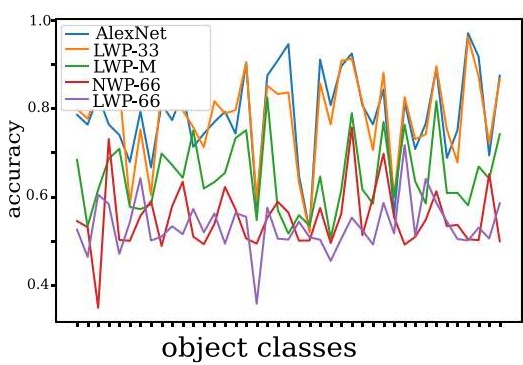

Figure 2: AlexNet average accuracy for different pruning rations $\theta$

Figure 3: Comparison of classification accuracy between unpruned AlexNet and pruned using NWP-M, LWP-33, NWP-66, LWP-66.

The experimental section is organized as follows:

- Sensitivity to pruning
- Study of the Filter Interference

## 3.1 SENSITIVITY TO PRUNING

We started our study by firstly evaluating the sensitivity of the single class classifier for pruning. In particular, we started with an evaluation of the discrete pruning approach. We applied it to both layer-wise (LWP) and network-wise pruning(NWP).

### 3.1.1 DISCRETE PRUNING

The process of semantic pruning starts by separating the dataset into subsets of images belonging to the selected object label. Let $m$ be the total number of filters in AlexNet. We calculated accumulated responses $\gamma$ for all $m$ filters in the network. Then we sorted the values in the descending order and applied four different settings to access the effect of discrete pruning:

- 0 score - removing all filters with $\gamma \leq 0$
- 0.33 - removing 33% of Filters with lowest value of $\gamma$
- 0.66 - removing 66% of Filters with lowest value of $\gamma$
- median (M) - Filters with $\gamma$ value lower than the median of all $\gamma$ were removed.

The procedure was repeated for all selected 50 classes separately.

Figure 2 shows the effect of discrete pruning on the number of parameters removed and preserved. Experiments indicated that pruning filters with NWP-0 and NWP-33 affect the average network accuracy by around 10% (Figure 2), while layer-wise pruning looks to be more resistant toward higher ratios. However, both the LWP-M and NWP-66/LWP-66 result in a serious drop of network accuracy (green, red and purple lines in Figure 3), and the difference between layer-wise and network-wise pruning are not as large as expected (red and purple lines).

Figure 2 also shows that removing even the least active layer-wise filters reduces the general network accuracy. This can be seen when comparing NWP-33 and LWP-33 in Figure 2 and 3 respectively. However, it can also be observed that the classification of certain classes becomes more accurate while other classes suffer much steeper loss in classification accuracy for LWP-33 (Figure 3 yellow curve).

Finally, note that no normalization of activity measure was used. The reason for not using normalization is due to the fact that we are evaluating pruning with respect to one class at a time. Therefore, normalizing the activity with respect to the whole data set would result in sample dependent

pruning and therefore remove the desired class wise pruning result. Additionally, normalizing with respect to one class at a time will introduce different minimal and maximal filter activities making a magnitude-based comparison between classes meaningless. Therefore, simply taking into account the NWP or LWP allows to account for both these effects and still provide reasonable information about the pruned and unpruned filters.

Figure 4 shows how the pruning affects the individual layers of filters in the feature extractor. Figure 4a shows the relative amount of pruned filters at ratio of 5%, Figure 4b at 25% and Figure 4c at 50%. The $x$ axis indicates the layer and $y$ axis the mount of filters.

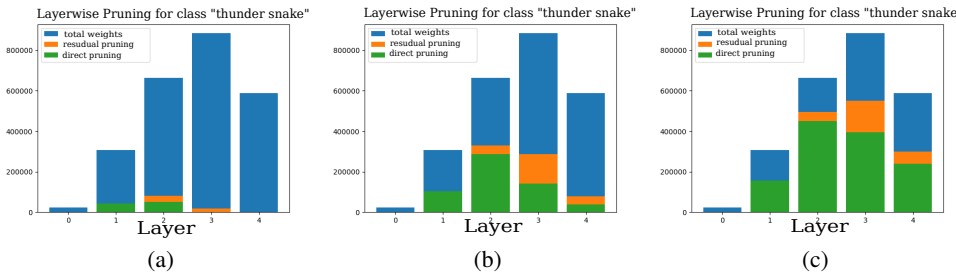

Figure 4: The relative amount of filter pruned by layer and by pruning ratio (a) $\theta = 0.05$, (b) $\theta = 0.25$, (c) $\theta = 0.5$

In Figure 4 the *Direct Pruning* (DP) represents the number of filters removed by the semantic pruning method. The *Residual Pruning* (RP) represents the pruning resulting from the *DP* pruning. The *RP* pruning represents the filters that became inactive on average because their input becomes 0 on average after the *DP* pruning. Observe that similarly to the previous experiment, filters in $0^{th}$ layer do not get pruned at all. Similarly, filters in the $4^{th}$ layer do not get pruned as well at low $\theta$. Filters from the $1^{st}$ and $2^{nd}$ layer is the one most pruned in DP while the filters of the $3^{rd}$ layer are pruned as a consequence in RP. Additionally, notice that at 50% of pruning ration almost all filters from the $2^{nd}$ layer are removed as a result of pruning. It is very interesting to see that even with higher pruning ratios, the $0^{th}$ layer is not pruned at all and the $4^{th}$ layer is pruned only at higher values of $\theta$. This means that the encoding provided by the first and last layer seems to be the most crucial and the densest.

### 3.1.2 CONTINUOUS PRUNING

The experiments with discrete pruning showed some important information about structural relationships of filters and the accuracy of classification of certain classes. However this information is only partial complete and therefore a continuous process of pruning was also evaluated with respect to individual class accuracies. In the continuous pruning, $\theta$ was set to 0 and was increased by 0.1 until reaching $\theta = 0.6$ (representing 60% pruning ratio). Figure 5 show the plot of class-wise accuracy for twelve selected object labels. It can be noticed that:

- Several classes are not affected by the pruning unless a very large ratio ($\theta \geq 0.15$) of filters is pruned (Figure 5e- 5h).
- Additionally, several classified classes has its accuracy increased after pruning. This is quite remarkable as this also entails that there can be interference between filters. (Figure 5a- 5d).

### 3.2 STUDY OF THE INTERFERENCE

Semantic pruning showed that pruning least active filters from one class can result in increased accuracy of classification. The possible explanation for the phenomena is filters interference within an overcrowded region of the feature space.

To investigate these phenomena in more details, we decided to investigate how many different and shared filters are pruned, and how the pruning affected the class density within the feature space.

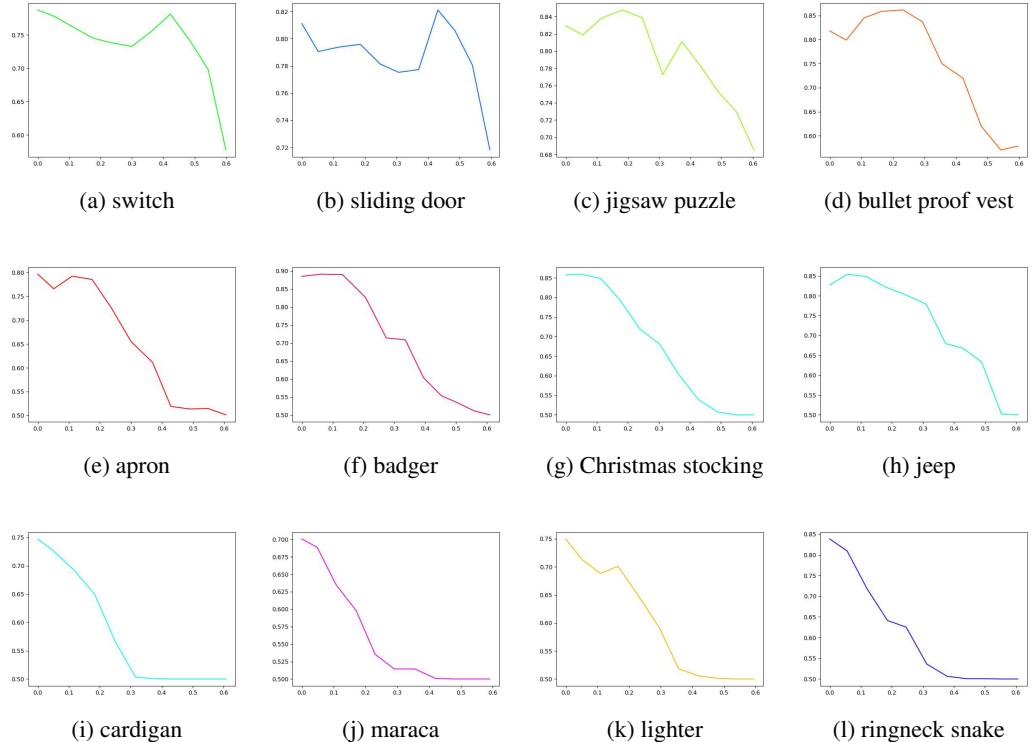

Figure 5: Examples of (a)-(d) object classes that improved for pruning ratio $\theta > 0$, (e)-(h) that for certain ratio $\theta > 0$ have same accuracy of classification as when $\theta = 0$ and (i)-(l) object classes for which classification accuracy was always worse than when classifying when $\theta = 0$. x-axis - $\theta$, y-axis - accuracy.

In order to determine the overlap between object classes, inter-class feature overlap was computed by calculating the intersection of unpruned $\mathbf{F}_\sigma$ and pruned $\mathbf{F}_{\bar\sigma}$ sets of filters. Let two object classes $\lambda_1$ and $\lambda_2$ be represented by $\mathbf{F}_\sigma^{\lambda_1}$, $\mathbf{F}_{\bar\sigma}^{\lambda_1}$ and $\mathbf{F}_\sigma^{\lambda_2}$, $\mathbf{F}_{\bar\sigma}^{\lambda_2}$ respectively. Then let $\rho_\sigma = \mathbf{F}_\sigma^{\lambda_1} \cap \mathbf{F}_\sigma^{\lambda_2}$ and $\bar\rho_\sigma = \mathbf{F}_{\bar\sigma}^{\lambda_1} \cap \mathbf{F}_{\bar\sigma}^{\lambda_2}$ be a double measure filter similarity. The reason for using both $\mathbf{F}_\sigma$ and $\mathbf{F}_{\bar\sigma}$ is to utilize two sources of information. In other words, we want to determine similarities of features which are required, and what features are not required by the object class. Higher is the value of $\mathbf{F}_\sigma$ higher the similarity of filters used for classification of $\lambda_1$ and $\lambda_2$.

Figure 6 provides the correlation between pruned and un-pruned filters for the three classes. (Only three classes were selected due to the space limitations.) All three classes represent different behaviour toward pruning: performance improvement, pruning resistant and accuracy degradation.

The general trend for all three graphs for non-pruned filters is that the correlation between filters is constantly decreasing, as one can expect. More interesting trends, however, can be observed for the correlation between pruned filters.

The first object class we would look at is *jigsaw puzzle*. According to Figure 5c, there is a significant effect of the pruning on the classification accuracy of the jigsaw puzzle class. Possible explanation can be provided by the fact that the classes-neighbours (Figure 6a and Figure 6d) for this class are semantically unrelated. Hence, the reduction in a correlation of pruned filters is reducing the interference with class-neighbours filters ($\theta \in \{0.075, 0.2\}$), which match the region of the sudden jump in accuracy according to Figure 5c. It is also can be noticed that for $\theta = 0.3$ semantically unrelated classes started to have increased in pruned-filter correlation to the *jigsaw puzzle* class, which reflected in the sudden accuracy drop in Figure 5c. A similar trend can be observed for another class *jeep*. At the region where $\theta \in \{0.05, 0.15\}$, we can see that correlation with more semantically related classes: *beach vagon, recreational vehicle*, is increasing, while *sax and jigsaw*

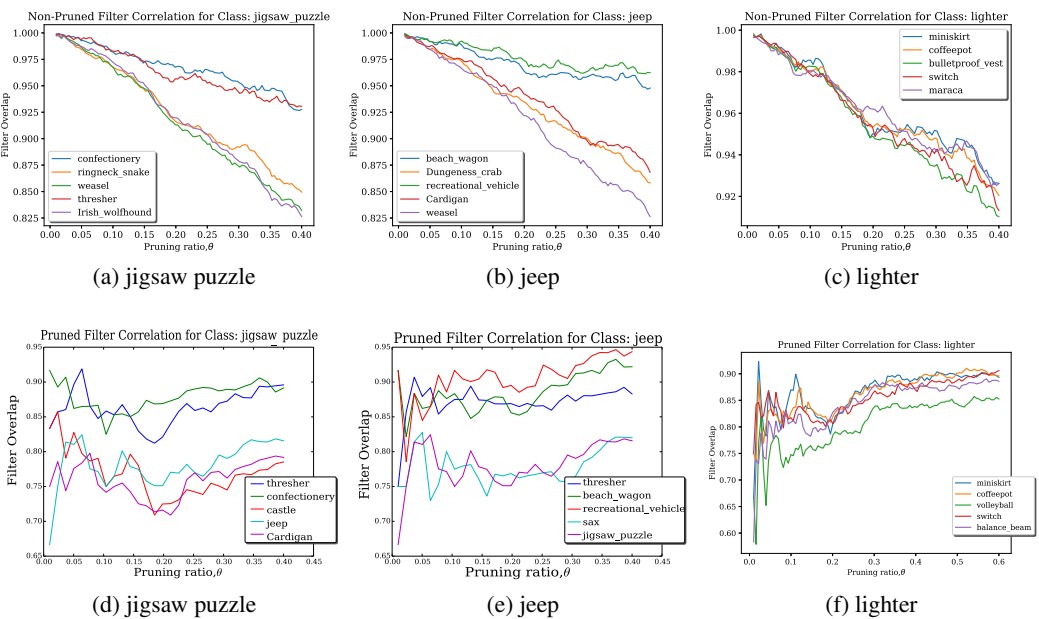

Figure 6: Correlation of filters (a)-(c) left after pruning, (d)-(f) removed after pruning

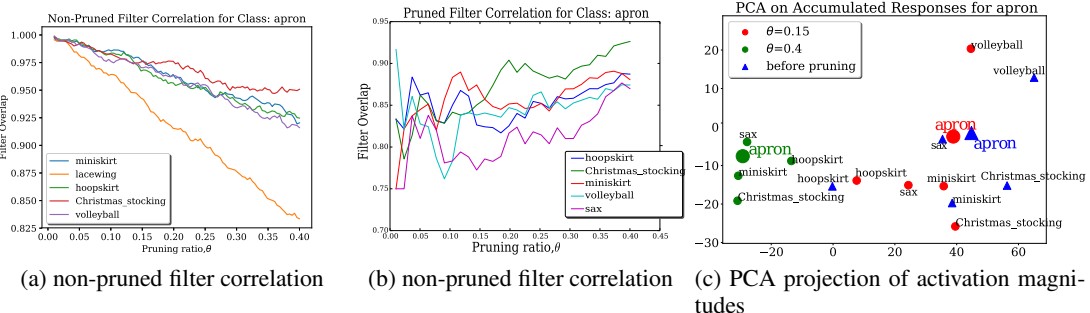

Figure 7: Correlation of filters left after pruning (a), removed by pruning (b) and PCA mapping of accumulated response for class *apron*

*puzzle* is decreasing. It is possible to assume that, the effect of interference from unrelated filters is reducing, and as a result, improves the model accuracy, as shown in Figure 5h. The last example is provided for the class from the group, which was negatively affected by the pruning (Figure 5k). Looking at Figure 6f we can see that **lighter** class has a high correlation for pruned filters with neighbouring classes; hence the effect of interference from the class-neighbours was not negated, which is reflected in the accuracy drop. However, one can observe that during the drop in pruned filter correlation for $\theta \in \{0.15, 0.21\}$ there is a sudden jump in classification accuracy (Figure 5k).

More extended evaluation is provided in Figure 7 for the class **apron**. According to Figure 7b, at the pruning rations $\theta = [0.1, 0.2]$ the correlation between pruned filters is increasing with such classes as *hoopskirt, Christmas stoking* and *miniskirt*, while the correlation is reducing for volleyball and sax. Semantically, the first three classes are more related to the class apron than the last 2. In such a way, at pruning ratio, $\theta = [0.1, 0.2]$ filters would be more associated with the related classes, which at the same time should positively affect the performance. That is what can be observed in Figure 5e. There is a performance jump in that region for $\theta = [0.1, 0.2]$. Further, the point can be supported by calculating the accumulated responses $\gamma^{apron} = \{\gamma_1^{apron}, \cdots, \gamma_q^{apron}\}$ for all filters in AlexNet and mapping them to 2D space through PCA dimensionality reduction. 2D projections

(Figure 7c) shows that at the pruning ratio 0.15 (which is in range of 0.1-0.2) class *apron* lay closer to the correlated classes, and moving away from unrelated "sax" and "volleyball" classes. While at the pruning ratio 0.4, it got moved to the unrelated classes again, which also affected the model accuracy.

The general observation from this experiment leads to the conclusion that the objects are represented in the feature space in areas of different density. Interestingly, the accuracy of classes was improving for the pruning ratios at which the correlation of pruned filters with semantically related classes was increasing, while the correlation with semantically unrelated filters was decreasing. These observations are supporting the hypothesis about filter interference and giving a promising direction for further investigation and potential classification accuracy improvements. Conducted experiments showed an interesting observation about the correlation between accuracy and class-wise similarity between pruned filters $\mathbf{F}_{\bar{\sigma}}$.

## 4 CONCLUSIONS

We presented a study of semantic CNN pruning. The results indicate that classification classes are asymmetrically represented by filters resulting in the fact that some object classes have their classification accuracy increased when pruned for. We also observed that objects classes represented in densely populated parts of the feature space share very similar filters. Such object classes when pruned for the least active filters (small $\gamma$) have a tendency to affect close neighbouring classes more directly and therefore, the classification accuracy of the pruned object increases. Additionally, we also observed that the filters in the lowest layer are rarely pruned, similarly to the first layer. Finally, we observed that there is an interference between filters affecting classification accuracy, and there is a correlation between accuracy and class-wise similarity between pruned filters. As future work, we plan to extend our work to more classes and more models. In addition, we will study the effect of filter overlap on convolutional networks safety and adversarial examples phenomena.

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
