# OpenReview forum: "Semantic Pruning for Single Class Interpretability"
_ICLR.cc/2020/Conference — Reject_

### Official Review · AnonReviewer2 · 2019-10-19
**Official Blind Review #2**

**Rating:** 3

**Review:**

This work proposes to prune filters in CNN model to interpret the correlation among different filters or classes. Though interpreting CNN filter is a well-studied topic, learning the interpretability through pruning is new and interesting. The proposed method is simple, by just using the averaging the value of the output of each filter as the indicator. The author claims that object classes represented in high feature density area usually share similar filters, which is in accordance with the common sense. Also, filters at lower layer are usually important.

Some questions:
1.	Since each filter is still like a black box, is it possible to visualize some result of the discovered interpretability?
2.	I’m confused with the implementation of pruning. If the filter at layer j-1 is pruned, then the dimensions of the filters at layer j should also change. How this issue is dealt with? Further, will this dimension reduction, instead of the pruned filter itself, influence the model performance?
3.	Why not use the absolute value of r_i? Any justification for this?
4.	The author mentioned that no normalization across categories are applied. However, are r_i from different layers comparable under NWP? Also, How can you guarantee that the Eq.(2) is comparable for different filters? More discussion on the normalization is desired.


**Experience Assessment:**

I have published one or two papers in this area.

**Review Assessment: Checking Correctness Of Derivations And Theory:**

I assessed the sensibility of the derivations and theory.

**Review Assessment: Checking Correctness Of Experiments:**

I assessed the sensibility of the experiments.

**Review Assessment: Thoroughness In Paper Reading:**

I read the paper at least twice and used my best judgement in assessing the paper.

---

> ### Author Response · Authors · 2019-11-11
> **Response to the Review #2  questions and comments**
>
> 1. Since each filter is still like a black box, is it possible to visualize some result of the discovered interpretability?
>
> Thank you for the suggestion. We also think this will improve the paper; however, due to shortage of time, we have not been able to include it in the article. But we definitely will include it in future drafts.
>
> 2. I’m confused with the implementation of pruning. If the filter at layer j-1 is pruned, then the dimensions of the filters at layer j should also change. How this issue is dealt with?
>
> That is a valid question. For simplicity, we substituted the pruned filters by zeros, so the dimensionality holds the same.
>
> 3. Why not use the absolute value of r_i? Any justification for this?
>
> Along with the notion of contribution, we wanted to measure the notion of unrelatedness of the filter to the selected class. Therefore we decided to use values before ReLU, and kept the sign of the value, instead of taking the absolute value.
>
> 4. The author mentioned that no normalization across categories is applied. However, are r_i from different layers comparable under NWP? Also, How can you guarantee that the Eq.(2) is comparable for different filters? More discussion on the normalization is desired.
>
> Thank you for this comment. That is one of the challenges we faced and need to be addressed in the future.
>
> The main challenge for doing the class-wise normalisation is that network response for each class is different, and, therefore, if we will be performing network-wise normalization, it can create a bias toward some classes, which can create problems with proper pruned filters comparison.
> Therefore, for most of the experiments, we used layer-wise pruning, unless otherwise specified, although we will investigate this point in more detail in our future work.
>
> Another reason that we did not use normalization is that we only compared classes within the scope of one particular class. This means that currently in this work, we were not interested in providing a common trend of overall activations, but, instead, we looked only at pairwise differences between activations within the class.  Finally, our pruning method is using the percentage of k lowest activations, and therefore the normalization would not have had much effect.
>
> Equation 2 calculates the accumulative response. Its magnitude will be different for various classes, but because we only use pairwise evaluation, it was not used as of yet. However, normalization is also a criterion that we plan to investigate in our future work.

---

### Official Review · AnonReviewer1 · 2019-10-23
**Official Blind Review #1**

**Rating:** 3

**Review:**


Summary
---

(motivation)
This paper proposes a new approach to pruning activations from neural networks,
but uses it to understand neural nets better rather than trying to make them
more efficient. It prunes filters per-class 1) to measure how sensitive image
classes are to pruning and 2) to measure how similar classes are by comparing
the filters they prune.

(approach)
Using only images from class c, CNN pre-activations are aggregated across spatial
dimensions and examples. This gives an average feature vector for that class.
T percent of neurons are pruned. Features with lower pre-activations
(possibly negative with large magnitude) are pruned first with successively
higher activations pruned later.
No re-training is performed.

(experiments)
Experiments use AlexNet and 50 of the 1000 ImageNet challenge classes to show:
1. Pruning filters results in decreased accuracy with smaller decreases in accuracy as the first and last few filters are pruned.
2. More filters from middle layers are pruned than are those from early and late layers.
3. After some filters have been pruned, sometimes pruning can increase accuracy.
4. I could not understand section 3.2.


Strengths
---

The proposed pruning method is simple and efficient.

I like the broad goal of understanding CNNs. We could use more papers that just focus on analysis like this one.

I think the idea of class conditional pruning is novel.


Weaknesses
---

# Major Weaknesses

I don't see why these results are significant. I do not find these results very surprising (see next comment) and I do not see why the community will find them useful.

* In particular, consider the conclusion. Sentences 2, 3, 5, and 6 seem to all be observations about what happened in the experiments. The conclusion should re-iterate the results, but it should also say why they are important. How does the work relate to the goals of the community at large? Which goals? Will this enable important new capabilities? What general concepts did we learn from this that we didn't know before?

Some of the positions in the paper could be more carefully considered. There are alternate explanations for many of these phenomena.

* Pruning smallest activations doesn't make sense to me. Typically redundant or un"important" activations are pruned because doing so has negligible impact on accuracy. The smallest activations are pruned here. Should the smallest activations be redundant or unimportant in some way? Does this rely on ReLU activations following the pre-activations? I think these values are not necessarily redundant and could be quite important (e.g., as measured by some saliency explanation like Integrated Gradients), but I could be wrong. It would be useful to provide a baseline which removes highest instead of lowest activations.

* Much of the surprise about figure 5 seems to be because it is not monotonic but it was expected to be monotonic. I agree that these curves should generally go down as theta increases, but I don't see why that relationship should be strict. I would be surprised if activations were not in some way dependent on one another. Furthermore, why does that dependence have to be interference? Couldn't it also be that some activations are complementary (and thus ineffective when only one is present)?

* Does Network Wise Pruning (NWP) favor more layers than others? It may be that some filters have higher Accumulated Responses per filter simply because there are more feature maps in the previous layer (thus more things summed up) and not because of what information they capture or their relationships with other filters. Does this happen? This could be an alternative to the following conclusion: "This means that the encoding provided by the first and last layer seems to be the most crucial and the densest."


# Other General Weaknesses:

* AlexNet is a rather old architecture to use for this analysis, so I can't be confident these methods or behaviors will generalize to other architectures. Does it hold for more modern architectures?


# Missing details / Points of confusion:

* The paper says the correlation from Figure 6 should be expected to decrease as theta increases. Why should this be expected?

* What exactly does Figure 6 measure? I think it's the ratio of the size of the intersection rho_sigma to the size of the union of the same two sets. However, the text calls the metric "correlation." This should be made clearer.


# Minor presentation weaknesses

Some parts of the notation/explanation don't make sense to me:
* "Let Lambda=... be the number of object classes in the dataset." But Lambda is defined as a set, not a numeral.
* The number p needs more context/subscripts as it depends on class c and filter i.
* "θ = [0, . . . , 1]" This defines theta as a finite set, but I think it's meant to be a number in the interval range (0, 1).
* Why is lambda_c needed instead of just using the index c to specify a class? The variable lambda doesn't seem to be any different than a class index.
* "let Fσ and Fσ ̄ be the set of unpruned and pruned filters for a given γ..." I don't see how this works. According to eq. 3 gamma depends on a particular filter i and class c, so the only filter F could contain is the ith one. (Later it became clear that F was only mean to class conditional, not filter conditional.)

Figure 5: These plots shoul all have the same y axis range. This would make them comparable and allow readers to much more easily compare trends across classes. Similar steps should be taken so the same range is used any time theta is plotted on the x axis.

Figure 3: I find it hard to get an overall ordering of the approaches in this figure because there is so much variance from class to class. It effectively conveys the variance, but I'd also like to know what the means across classes are for each method so I can compare the proposed approaches more effectively.

"The results indicate that classification classes are asymmetrically represented by filters resulting in the fact that some object classes have their classification accuracy increased when pruned for."
* I'm not sure what it means to be asymmetrically represented by filters.


Suggestions
---

This analysis would have been more interesting with an existing pruning approach because we would already know that such an approach is good a removing unnecessary filters.

Final Evaluation
---

Quality: Experiments could have been cleaner, but they basically demonstrate the patterns the paper intended to show.
Clarity: I could understand most experiments at a high level, but I found it hard to understand the motivation and the rest of the experiments.
Significance: As explained above, I do not see why the paper is significant.
Originality: The experiments and the proposed class conditional pruning approach are somewhat novel.

The paper is somewhat novel, but do not find it very clear and I do not see why it is important, so I cannot reccomend it for acceptance.


**Experience Assessment:**

I have read many papers in this area.

**Review Assessment: Checking Correctness Of Derivations And Theory:**

I assessed the sensibility of the derivations and theory.

**Review Assessment: Checking Correctness Of Experiments:**

I assessed the sensibility of the experiments.

**Review Assessment: Thoroughness In Paper Reading:**

I read the paper at least twice and used my best judgement in assessing the paper.

---

> ### Author Response · Authors · 2019-11-11
> **Response to the Review #1 questions and comments. Part 1**
>
> 1. I don't see why these results are significant. I do not find these results very surprising (see next comment), and I do not see why the community will find them useful.
>
> We apologies for not being clear on the contribution of the work.
> Our main findings are the following: the accuracy of the classification of some classes was improved at specific pruning ratios. It turns out that this effect was observed when the correlation between the pruned filters (removed filters) of the pruned class and the pruned filters of semantically related classes was increasing, while the correlation with semantically unrelated classes was decreasing. These observations are supporting the hypothesis about filter interference and giving a promising direction for further investigation.
> The significance of the work resides in the fact that the class-wise pruning can be seen as a base for designing class-wise classifiers. By identifying the closeness of objects in a feature space,  one can use this information to develop classifiers that would be more accurate and will be classified only a set of well-selected object classes. Direct applications can be attributes prediction (multi-class classification), scene classification, etc.
>
> 2. In particular, consider the conclusion. Sentences 2, 3, 5, and 6 seem to all be observations about what happened in the experiments. The conclusion should reiterate the results, but it should also say why they are important.
>
> We apologies for not being clear in conclusion, about the possible contributions to the community. The correct achievements mentioned in conclusion should be as follows:
> In general approaches to pruning, people are looking at the filters which are left in the network after pruning (unpruned filters). However, we investigated the correlation between both pruned and unpruned filters. The most surprising result is that more exciting and quantifiable trends are found in the correlation of information the network is not using and, as a result, being pruned.
> The accuracy of the classification of some classes was improved at certain pruning ratios. It turns out that this effect was observed when the correlation between the pruned filters (removed filters) of the pruned class and the pruned filters of semantically related classes was increasing, while the correlation with semantically unrelated classes was decreasing. These observations are supporting the hypothesis about filter interference and giving a promising direction for further investigation and potential classification accuracy improvements.
>
> 3-5. How does the work relate to the goals of the community at large? Which goals?  Will this enable important new capabilities?
>
> For instance, you want to train an attribute predictor, which contains many different objects. By using class-wise analysis, we can separate these attributes in groups, which does not create interference for each other, which should lead to improving the accuracy of classification.
>
> 6. What general concepts did we learn from this that we didn't know before?
>
> The pruned information is more informative than the unpruned information. This is a novel observation as most of the already existing pruning methods do not use this information.
>
> 7. Some of the positions in the paper could be more carefully considered. There are alternate explanations for many of these phenomena.
>
> 	Yes, the reviewer is right; we address individual comments below.
>
> 8.  Pruning smallest activations don’t make sense to me. Typically redundant or un"important" activations are pruned because doing so has a negligible impact on accuracy. The smallest activations are pruned here. Should the smallest activations be redundant or unimportant in some way?
>
> We apologise for an unclear explanation of our reasoning for pruning the lowest activation. While what the reviewer mentions can be true for a general case pruning, the main goal of our work is to determine the filters which are most contributing to the selected class classification. So we are interested in determining filters which provide a high response to the images of the selected class when such are provided. In such a way, we want to identify class-specific filters.  In particular, we are collecting each individual class statistics (average activation response for all images for the selected class) and therefore pruning the least contributing activations is intended to preserve only filters that directly contribute to each class classification. The surprising result was that the accuracy was increasing when these filters were shared by two semantically very close classes.

---

> > ### Author Response · Authors · 2019-11-11
> > **Response to the Review #1 questions and comments. Part 2**
> >
> > 9. Does this rely on ReLU activations following the pre-activation?
> >
> > We are using features before ReLU because using ReLU will negate some of the important information provided by the activations. In other words, we want to determine the degree to what filters are unrelated to the class, rather than simply determining if the filter contributes to the class recognition or not.
> > 10. I think these values are not necessarily redundant and could be quite important
> > We also support this point of view, therefore we are nor using features after ReLU, which creates many zero values, which are not too much information for our purpose of determining the different degrees of unrelatedness.
> >
> > 11. It would be useful to provide a baseline which removes the highest instead of lowest activations.
> >
> > Thank you for the suggestions, we will address this in our future experiments. This point is partially addressed by the fact that we are measuring the correlation between both pruned and unpruned filters. And the similarity between pruned filters can be seen as if only high-activation neutrons were removed. But, that is a valid point to address in future work, thank you for the comment.
> >
> > 12. Much of the surprise about figure 5 seems to be because it is not monotonic but it was expected to be monotonic. I agree that these curves should generally go down as theta increases, but I don't see why that relationship should be strict. I would be surprised if activations were not in some way dependent on one another. Furthermore, why does that dependence have to be interference? Couldn't it also be that some activations are complimentary (and thus ineffective when only one is present)?
> >
> > The review is right, most of the previous works showed that with higher pruning ratio the dependency is monotonic, but they mainly looked at the overall model performance, rather than class-wise accuracies. However, having a monotonic trend on the global level, will not guarantee similar behaviour from each class separately (This can be seen in Figure 2, where the accuracy decreases monotonically, while for many of the Figures in Figure 5 the behaviour is not monotonic). Therefore, we decided to make a further investigation.
> >
> > If the activations would be only complementary, the pruning of a complementary filter would only decrease the class-wise accuracy. If however an interfering filter (even from a set of related or complementary filters) would be removed, an increase of accuracy should occur (Figure 5 a-d).
> >
> >
> > 13. Does Network Wise Pruning (NWP) favour more layers than others? It may be that some filters have higher Accumulated Responses per filter simply because there are more feature maps in the previous layer (thus more things summed up) and not because of what information they capture or their relationships with other filters. Does this happen? This could be an alternative to the following conclusion: "This means that the encoding provided by the first and last layer seems to be the most crucial and the densest."
> >
> > That is a valid point. Another reason why activation of intermediate layers can be smaller is due to high sparsity of the input activation maps. Therefore, for most of the experiments, we used layer-wise pruning, unless otherwise specified. Although, we will investigate this point in more detail in our future work.  One of the arguments to support our conclusion can be drawn from the multiple works in network binarization, where it is accepted not to binaries the first and the last layers.
> >
> >
> > 14. AlexNet is a rather old architecture to use for this analysis, so I can't be confident these methods or behaviours will generalize to other architectures. Does it hold for more modern architectures?
> >
> > While the AlexNet is a relatively old model, however, it is still a valid model for investigation. We agree that more models should be evaluated. By the time of the paper submission, the results were not ready. Currently, we can observe similar behaviour for VGG model as well. Further experiments on ResNet are also in progress.

---

> ### Author Response · Authors · 2019-11-11
> **Response to the Review #1 questions and comments. Part 3**
>
> 15. The paper says the correlation from Figure 6 should be expected to decrease as theta increases. Why should this be expected? What exactly does Figure 6 measure? I think it's the ratio of the size of the intersection rho_sigma to the size of the union of the same two sets. However, the text calls the metric "correlation." This should be made clearer.
>
> We apologise for not providing enough details about this point.
> For these figures, the y-axis (Filter overlap) represent the intersection between either unpruned (a-c) or pruned (d-f) filters. It was calculated in the following manner:
> 1. Each class will have an n-dimensional vector, where n is a number of filters in the network.
> 2. Cosine similarity is calculated between vectors of different classes, which will be in the range [0,1]
>
> The similarity value from Figure 6 (a-c) should be decreased as theta increases, as more and more filters are getting pruned. As a result, fewer and fewer filters will be left. While in figures (d-f), the opposite trend should be observed. As more filters are pruned, hence more common pruned filters classes will have.
>
> 16.  Minor presentation weaknesses
>
> Thank you for the observation; we will simplify notations in the next version of the drafts.
>
> 17. Figure 5: These plots should all have the same y-axis range. This would make them comparable and allow readers more easily compare trends across classes. Similar steps should be taken, so the same range is used any time theta is plotted on the x-axis.
>
> Thank you for these comments; the scales on the plots were selected automatically for better visualisation. But we will consider the unification of the scale.
>
> 18. Figure 3: I find it hard to get an overall ordering of the approaches in this figure because there is so much variance from class to class. It effectively conveys the variance, but I'd also like to know what the means across classes are for each method so I can compare the proposed approaches more effectively.
>
> We apologise for the unfortunate Figure 3; we will improve the visualisation in the future. Figure 2 shows similar trends, but provides the mean value for all classes, instead of giving the class-wise values separately.
>
> 19. "The results indicate that classification classes are asymmetrically represented by filters resulting in the fact that some object classes have their classification accuracy increased when pruned for.” * I'm not sure what it means to be asymmetrically represented by filters.
>
> Sorry for the confusing formulation. What we intended to say is the following:
> Not all filters contribute positively to the class classification and can create interference for the class recognition.
>
> 20.  This analysis would have been more interesting with an existing pruning approach because we would already know that such an approach is good at removing unnecessary filters.
>
> As the reviewer pointed out correctly, there are much more efficient approaches that exist and do a much better job; however, the target of the paper is not pruning as a method for optimization, but rather the analysis of class-wise filter contribution.  Besides, some of the existing pruning methods are not applicable for our purposes, as they are optimized for different objectives. For example, pruning based on weights values, are not appropriate for our case, as network parameters are shared for all classes.
> Accuracy or loss based optimizations will be challenging to use as well, as we want to investigate the original filter contribution to an individual class classification, and do not want to affect the model accuracy by applying any retraining.
> We selected to use this straightforward pruning because it allows us to have the highest control on what filters get removed.
>
> However, it is a valid point, and we will consider alternative pruning methods in the future, to verify if our funding still would hold there. Thank you for the suggestion.
>
> 21.  I could understand most experiments at a high level, but I found it hard to understand the motivation and the rest of the experiments.
>
> We are sorry that we didn’t provide a more precise description of the motivation.
> The primary motivation of the work is to lay the ground for the optimisation procedure for learning:
> Intelligently separate classifiers based on the interclass interference should allow training better classifiers.  More details for the motivation have been provided to question 1 and 5.

---

### Official Review · AnonReviewer3 · 2019-10-23
**Official Blind Review #3**

**Rating:** 1

**Review:**

This paper proposes to prune CNN networks for each class in order to study interpretability of the networks. However, there are several significant drawbacks:

1) The pruning approach is too simplistic. Network pruning has been a field of very active research as the authors have acknowledged, however, the approach used in the paper is a very simplistic remove the ones with lowest response one, for which, there is no experiment to justify its validity w.r.t. state-of-the-art pruning approaches. In fact, from Fig. 2 and Fig. 3 one can almost conclude that this naive pruning method is significantly inferior to state-of-the-art.
2) Lack of novel insights. For an explanation paper, we would expect to obtain some new insights about the classification. The results that this paper get to, such as few filters were pruned in the lower layers, similar classes share similar filters, are not necessarily new knowledge to the community. In terms of the result analysis, mostly simple correlation analysis was used which presented no novelty nor insights.

**Experience Assessment:**

I have published one or two papers in this area.

**Review Assessment: Checking Correctness Of Derivations And Theory:**

I assessed the sensibility of the derivations and theory.

**Review Assessment: Checking Correctness Of Experiments:**

I assessed the sensibility of the experiments.

**Review Assessment: Thoroughness In Paper Reading:**

I made a quick assessment of this paper.

---

> ### Author Response · Authors · 2019-11-11
> **Response to the Review #3 questions and comments**
>
> 1) The pruning approach is too simplistic.
>
> The main objective of the paper is to study and analyze the relationship of the learned filters in CNN, rather than to propose a new pruning method.
> We selected this simple pruning approach to avoid unnecessary complexity of pruning criteria that would obfuscate the relations between learning objects. While more complex pruning methods would probably achieve better pruning ratios, as correctly mentioned by the reviewer,  this simple method allows looking at every learned object class on a neuron-wise fashion.
>
> Currently, available methods can be split into three main groups:
> - pruning based on weights values, which is not appropriate for our case, as network parameters are shared for all classes.
> - activation based methods. This line of work is the closest to our approach, as it focuses on removing the least active neurons. One of such works is [Luo 2017]. However, this method is not applicable in our case, as the authors are removing filters that do not change the activation map. Hence, such criteria do not provide sufficient information about if the particular filter is important for a specific class or not.
> - accuracy or loss based optimizations. This line of work prune and optimize neural network based on the contribution of the filter parameters to the overall loss function or overall model accuracy. Such models usually involve a training component. For instance, the Taylor pruning from [Molchanov 2019] is much more efficient, but it is optimized towards the final loss function and for additional retraining. Both of these criteria are not suitable for us, as we want to investigate the original filter contribution to an individual class classification. Besides, we do not want to affect the model accuracy by applying any retraining.
>
> Hence, the reason for selecting this - one of the simplest - pruning methods was motivated by the necessity of an approach that would directly allow us to prune the neurons directly related to a particular classified object class, to oppose to the pruning method which is based on the overall data statistics.
>
> 1. P. Molchanov, A. Mallya, S. Tyree, I. Frosio, and J. Kautz. Importance Estimation for Neural Network Pruning. CVPR, 2019.
> 2. J.-H. Luo, J. Wu, and W. Lin. Thinet: A filter level pruning method for deep neural network compression. ICCV, 2017.
>
>
> 2) Lack of novel insights.
> Main findings:
> As the reviewer pointed correctly, the points “the few filters were pruned in the lower layers, similar classes share similar filters” are not new discoveries. These points were just reported to provide a complete picture of the analysis.
>
> The main contribution, however,  is a discovered dependency in the correlation of pruned filters:
> - In general, people are looking at the filters which are left in the network after pruning (unpruned filters). However, we investigated the correlation between both pruned and unpruned filters. And more interesting trends are found in the correlation of information the network is not using and, as a result,  being pruned.
> - The accuracy of the classification of some classes was improved at certain pruning ratios. It turns out that this effect was observed when the correlation between the pruned filters (removed filters) of the pruned class and the pruned filters of semantically related classes was increasing, while the correlation with semantically unrelated classes was decreasing. These observations are supporting the hypothesis about filter interference and giving a promising direction for further investigation and potential classification accuracy improvements.
>
>
> On simple correlation usage:
> We used a simple correlation because we wanted to see the relation between the classes in a neural network. The simple correlation allows us to relate the magnitude of the network parameters to each classified class.  Besides, simple correlation measurement provides us with fast results, in alternative to more advanced similarity measurement methods.

---

### Decision · Program_Chairs · 2019-12-19

**Decision:**

Reject

**Comment:**

The authors propose to use pruning to study/interpret learned CNNs. The reviewers believed the results were not surprising and/or had no practical relevance. Unlike in many cases, two of the reviewers acknowledged reading the rebuttals, but were unswayed.